# Energy Efficient Graph-Based Hybrid Learning for Speech Emotion Recognition on Humanoid Robot

**Haowen Wu [1], Hanyue Xu [1,2] , Kah Phooi Seng [1,3,4,*], Jieli Chen [1,2] and Li Minn Ang [4]**

[1] School of AI and Advanced Computing, Xi'an Jiaotong-Liverpool University, Suzhou 215000, China; haowen.wu20@student.xjtlu.edu.cn (H.W.); hanyue.xu19@student.xjtlu.edu.cn (H.X.); jieli.chen22@student.xjtlu.edu.cn (J.C.)

[2] Department of Electrical Engineering and Electronics, University of Liverpool, Liverpool L69 3GJ, UK

[3] School of Computer Science, Queensland University of Technology, Brisbane, QLD 4000, Australia

[4] School of Science, Technology and Engineering, University of the Sunshine Coast, Petrie, QLD 4502, Australia; lang@usc.edu.au

\* Correspondence: jasmine.seng@xjtlu.edu.cn

**Abstract:** This paper presents a novel deep graph-based learning technique for speech emotion recognition which has been specifically tailored for energy efficient deployment within humanoid robots. Our methodology represents a fusion of scalable graph representations, rooted in the foundational principles of graph signal processing theories. By delving into the utilization of cycle or line graphs as fundamental constituents shaping a robust Graph Convolution Network (GCN)-based architecture, we propose an approach which allows the capture of relationships between speech signals to decode intricate emotional patterns and responses. Our methodology is validated and benchmarked against established databases such as IEMOCAP and MSP-IMPROV. Our model outperforms standard GCNs and prevalent deep graph architectures, demonstrating performance levels that align with state-of-the-art methodologies. Notably, our model achieves this feat while significantly reducing the number of learnable parameters, thereby increasing computational efficiency and bolstering its suitability for resource-constrained environments. This proposed energy-efficient graph-based hybrid learning methodology is applied towards multimodal emotion recognition within humanoid robots. Its capacity to deliver competitive performance while streamlining computational complexity and energy efficiency represents a novel approach in evolving emotion recognition systems, catering to diverse real-world applications where precision in emotion recognition within humanoid robots stands as a pivotal requisite.

**Keywords:** energy efficient deep learning; graph convolutional neural network; speech emotion recognition; humanoid robot

## 1. Introduction

As the first attribute of language, speech plays a decisive supporting role in language, including not only the text content that the speaker wants to express, but also the emotional information that the speaker wants to express. With the continuous expansion of social media, human–computer interaction and other fields, emotion recognition has gradually become one of the core technologies to achieve more natural and intelligent human–computer interaction [1] and social intelligence [2]. However, speech emotion recognition (SER) is a difficult task in a human–computer interaction system, because human language emotion change is an abstract dynamic process, and it is difficult to describe its emotional interaction with static information.

With the wide application of deep learning, the task of speech emotion recognition has also attained better achievements. Before that, researchers usually extracted acoustic features expressing emotions from the collected speech signals, and found out the mapping

relationship between these features and human emotions to complete SER. The representative technologies include the Hidden Markov model (HMM) [3], support vector machine (SVM) [4] and Fourier parameter model (FPM) [5]. These works often require a manual design and selection of features, while deep learning models can automatically learn abstract high-level features from raw data, which allows the model to reduce the omission of helpful information for emotion classification. Many researchers have made excellent improvements [6–10] in speech emotion recognition using deep learning models, such as convolutional neural networks (CNN), Long Short-Term Memory (LSTM), and recurrent neural network (RNN). CNN are deep learning models specifically designed to process structured data, such as images, text, and audio. The core idea is to extract the features of input data layer by layer through convolution layer, pooling layer and fully connected layer to realize the learning of complex patterns. The excellent performance of CNN is mainly due to the weight sharing and local perception mechanism of the convolutional layer, which makes CNN perform well in image classification [11], object detection [12] and other tasks. In terms of emotion recognition, CNNS can effectively capture local speech features and retain their timing information, thus providing a powerful feature representation for emotion expression. For instance, Bertero [13] first studied speech emotion recognition based on CNN and successfully predicted emotions with 66.1% accuracy. Based on this work, Anvarjon [14] developed a lightweight speech emotion recognition system based on CNN, which improves the emotion discrimination performance of SER systems by utilizing ordinary rectangular filters. Moreover, Li et al. [8] proposed a SER framework that blends the CNN model and the LSTM model and classifies speech emotions through end-to-end context awareness. However, due to the characteristics of CNN, it cannot clearly model speech dynamics. The dynamic change in emotion will be affected by the dynamic of time, but CNN lacks in capturing the dynamic characteristics of time [15].

Graph convolutional networks are a class of deep learning models especially de-signed for processing graph data (such as social networks [16], recommendation systems [17], etc.). Unlike traditional neural networks, GCN can effectively capture the relationship between nodes and the structure information of graphs. Through the layer-by-layer propagation of the graph convolution layer, GCN is able to learn the node representation and achieve effective feature extraction in the graph data [18]. In tasks such as emotion recognition, speech data can be represented as a time sequence diagram, in which each node contains speech features, and each node is connected by edges to represent the dependency relationship between speech features. This data representation can better capture the complex structure and relationship of speech data. GCN is able to take into account node relationships in the data to more fully understand and analyze complex association patterns. For example, Kim and Kim [19] proposed a graph structure based on cosine similarity to represent speech data and used graph neural networks for expression learning of speech emotion recognition. Moreover, Amir and Tanaya [20] proposed a depth graph learning method that utilized GCN to learn speech features and perform emotion classification by modeling speech signals as graph data. Based on GCN, Li et al. [21] introduced an LSTM aggregator and weighted pooling approach to achieve better performance than existing graph learning baselines. Although GCN captures the sequence information lost by CNN in the convolution process, the performance of GCN is not as good as that of CNN in local feature extraction.

Although deep learning has made breakthrough progress in speech and emotion recognition, the computational and storage resources required grow with the improvement of model performance. The need for high computing power leads to higher carbon emissions and discourages participation by small and medium-sized companies and research institutions with limited funding, thus undermining equity in research [22]. With the continuous improvement of hardware computing power, deep learning can achieve quite good performance on a variety of tasks. Reducing the resource consumption of deep learning under the premise of ensuring performance has become a very important link at present. To capture the temporal dynamics of information, the most common model used by researchers

is the LSTM. Although RNN models such as LSTM already have good performance, it is usually a complex architecture with millions of training parameters, and this energy consumption cannot be ignored [23]. Therefore, we propose an energy efficient graph-based hybrid learning model for speech emotion recognition in a humanoid robot. It can not only capture the local features of speech data through CNN model, but also enhance the extraction of time dynamic information. In addition, since the speech data are constructed as graph data to strengthen the relationship between the speech signals, the training parameters similar to those required by LSTM for speech timing capture are also reduced. With the rapid development of artificial intelligence and robot technology, the humanoid robot has gradually become one of the research hotspots. With flexible limbs and a head, they are endowed with the ability to interact with humans. As a key part of robotics, emotion recognition [24] aims to give machines the ability to sense and understand human emotions in order to better adapt to and serve human society. By simulating human emotional expression, humanoid robots are able to respond more sensitively and intelligently to user needs in interactions.

In human–computer interaction, the introduction of emotion recognition by humanoid robots marks a new paradigm of human–machine collaboration, in which machines not only perform tasks but also interact with users on an emotional level. In 2020, many issues in the field of robotics and emotion recognition have emerged at this intersection, attracting the attention of many researchers [1]. For example, Dwijayanti [25] realized the feat of real-time facial recognition of humanoid robots to complete its emotion recognition by using the CNN model in 2022. This success greatly strengthens the robot's more human-like performance in human–computer interaction, and also lays the foundation for other modes and even multi-modal emotion recognition in the future. In 2023, when large models and Chat-GPT are popular around the world, people's in-depth thinking of the BERT model enabled Mishra [26] to realize real-time emotion generation of robots by using human–machine dialogue. Similarly, although real-time interaction has not yet been realized, the multi-modal emotion recognition model proposed by Hong [27] in 2022 is equally wonderful. Based on social robots, the human–robot interaction proposed by Hong is capable of human-like emotional feedback and actions on human emotions, thus giving more positive feedback to human beings. The contributions of this paper can be summarized as follows:

1. A fusion model combining CNN and GCN for audio emotion recognition is proposed, and the advantages of two commonly used emotion recognition in time series and local features are combined to enhance the capture of human audio details and improve the accuracy of emotion recognition.
2. The fusion model saves a lot of convolution processes of the CNN model, reduces the computing power resources of the GCN model to construct the graph data structure, to realizes part of the energy efficiency.
3. Realization of the emotion recognition of human–computer interaction. After the Nao robot is connected to the host to train the model through a specific SDK, it can judge human emotion problems with high accuracy according to human voice, so as to further expand the feasibility of robot home teaching or the teaching of the young age.

## 2. Literature Review

Before the field of deep learning, people mostly relied on traditional machine learning algorithms, such as the Random Forest model [28], support vector machine model [4], Gaussian mixture model [29], Hidden Markov model [3], etc., to identify emotions. However, the early model is not yet perfect and has great limitations in capturing the relationship of emotional data. With the advent of deep learning, CNNs [6–10] revolutionized the field of image processing, and early CNN-based models extracted layered features from facial images to identify basic emotions. At the same time, the RNNs [23] model with LSTM unit is also very helpful in establishing the time series model, which is used to capture the time–dependence relationship in speech signal and text data, and further realize emo-

tion analysis. And with the development of deep learning architectures, a deeper level of DCNNs models [11] followed, such as VGGNet and ResNet [30], which showed better performance in image emotion recognition. GNNs is followed by the establishment of graph structure data nodes to capture the relationship before the painless data modes for emotion recognition. When single-mode emotion recognition is relatively perfect, transformer architecture, such as BERT [31], has been proposed in the field of NLP to realize modal integration. In general, the history of emotion recognition has evolved from the early days of CNNs and RNNs to the most advanced multimodal fusion models and Transformer architectures. Continued research efforts in architecture design, multimodal integration, and self-supervised learning are expected to further improve the accuracy and robustness of emotion recognition systems.

In the aspect of sound emotion recognition, people's early research mainly focused on extracting features from sound signals, such as frequency, amplitude, etc., and capturing the emotional information of speech through these fundamental frequency changes. The resulting Mel-spectrogram [32] uses a Mel filter bank to simulate the human auditory system's perception of frequency and reveals emotional information by converting audio signals to it. A further implementation is MFCCs [33], which capture spectral features for emotion recognition through a combination of Mayer filters and discrete cosine transforms. Following the rise of deep learning techniques, DNNs models are able to extract features directly from raw sound data through end-to-end learning, enabling more accurate sentiment classification.

## 3. Methodology

### 3.1. Hybrid Emotion Recognition Network (HybridEmoNet)

3.1.1. Framework

In order to combine the advantages of CNNs and GCNs in audio emotion recognition tasks while reducing resource and memory consumption, we propose a hybrid model to apply it to humanoid robots that lack resources. The proposed model integrates CNN and GCN for audio and image emotion recognition. The CNN model can effectively simulate temporal and spatial cues, but when using real human data for analysis, due to the lack of information or weak correlation of network structure, the overall fitting of the model is lacking. Although the shared adjacency matrix and sparse connections of the GCN model strengthen their close connections among structural networks, they are inefficient in modeling spatio-temporal cues. To address these limitations, we build the HybridEmoNet, which aims to mitigate the shortcomings of CNN and GCN in order to enhance the extraction of spatio-temporal information of speech, and at the same time to achieve the purpose of energy saving.

Figure 1 illustrates the architecture of EmoNet, where the MFCC features of the audio are input into the CNN model and the spectral architecture sequence of the audio is placed into the GCN model. In the two trained models, we extract the unpooled feature vector of the original dimension from their convolutional layers, and then introduce the Fused Feature Extraction Module for feature extraction and fusion. Combining various aspects of graph-based and convolution feature extraction, it simultaneously captures structural information and spatio-temporal patterns that are highly characteristic of both models.

In the upper section of Figure 1, the architecture components of the CNN-only model are depicted. We obtained a Mel-spectrogram by using the short time Fourier transform (STFT) to process the audio signals. Then, we transformed it to MFCCs, which was obtained by the logarithmic processing of the spectrogram and DCT transformation to extract audio features. Following spectrogram preprocessing and expansion, the model underwent training. In the lower section of Figure 1, the architecture components of the GCN-only model are illustrated. By constructing a classification model of the graph structure, we assign a discrete emotion label to each transformed speech sample. We segmented the speech signals and set them as nodes and computed the corresponding adjacency matrix by calculating the correlation of the speech segments for the construction of the speech graph.

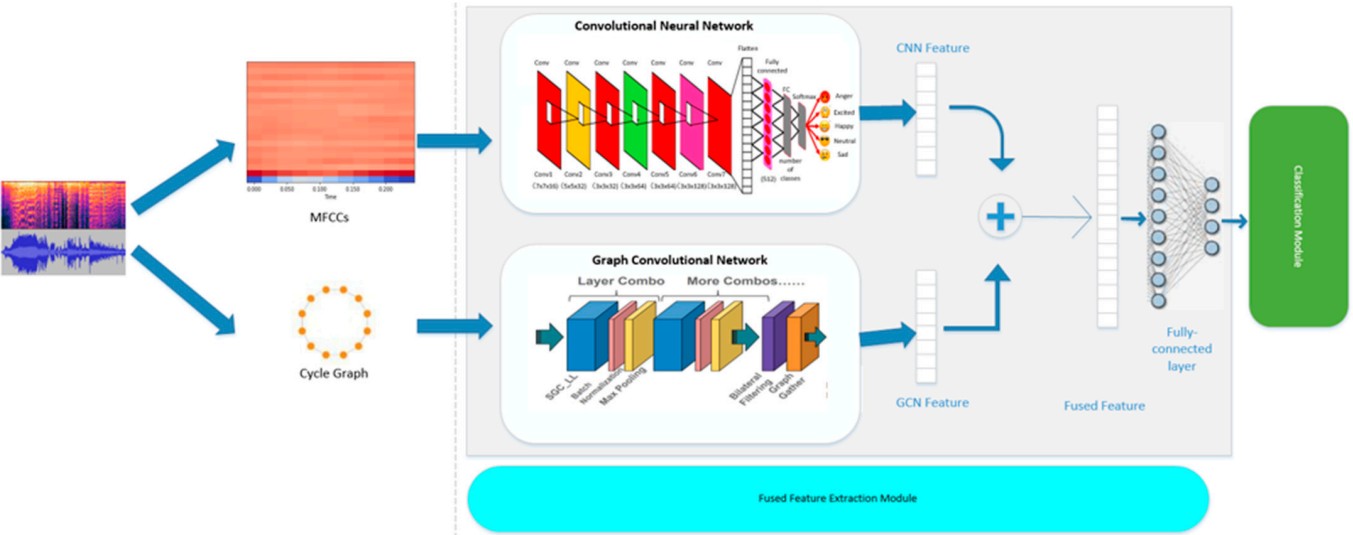

**Figure 1.** Framework of Hybrid Emotion Recognition Network.

The proposed Fusion Module (Fusion Module) fuses the data of two models and the fusion results are transmitted through self-focusing and adaptive connectivity layers. Subsequently, the fused data undergoes normalization and classification via a linear classifier. Discarded data using normalized scaled data are used to rewrite the lost data randomly.

### 3.1.2. Hybrid Model Fusion

The Fusion Feature Extraction Module (FFEM) in HybridEmoNet which is shown in Figure 2 is a key component designed to go beyond the inherent limitations of independent CNNs and GCNs. There is a piece of audio $x_i$ in dataset D. The input $x_i^{CNN}$ suitable for CNN and input $x_i^{GCN}$ suitable for GCN are obtained after data pre-processing. The output of the FFEM can be expressed as:

$$FFEM_{output} = W_g * x_i^{GCN} \oplus W_c * x_i^{CNN},$$

where $W_g$ and $W_c$ are the learnable parameters for graph and convolutional filters, respectively. The operator $\oplus$ denotes the concatenate operation, which indicates that the vectors obtained from CNN and GCN are tiled into one-dimensional vectors, respectively, and simply fusion is performed in Softmax layer as our feature concatenate. We adopt cross entropy as our loss function:

$$H(p,q) = -\sum_{i=1}^{n} p(x_i) log q(x_i)$$

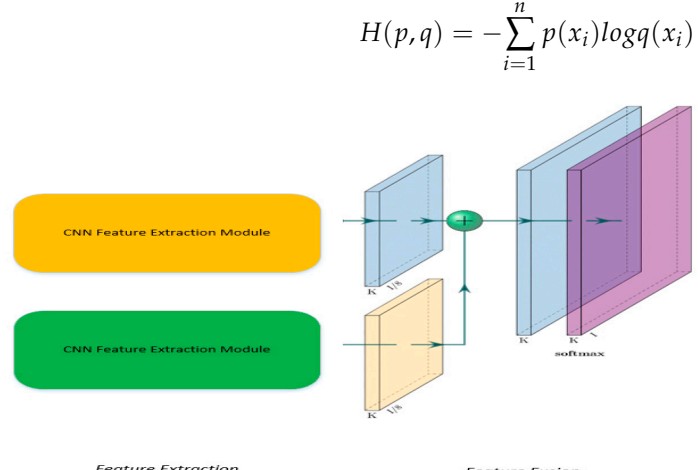

**Figure 2.** Framework of Fusion Extraction Module.

A distinctive feature of FFEM is its adaptive feature integration method, which dynamically adjusts the weights of the features extracted by the hybrid filter, maintaining a delicate balance between structural and spatio-temporal aspects. This adaptive capability empowers FFEM to efficiently discern and learn the various complex patterns present in the input data, resulting in a more discriminative feature representation.

### 3.2. Convolutional Neural Network

The Mel-Frequency Cepstral Coefficient (MFCC) is a fundamental element in the field of deep audio analysis and plays a key role in converting raw audio signals into concise and meaningful representations. This section explores in depth the MFCC extraction and preprocessing steps, combined with CNN.

### 3.2.1. Pre-Processing Procedure

In order to feed the audio into the CNN in the form of MFCC, first, we perform a framing step to divide the audio waveform into short windows:

$$x[n, m] = w[n] \cdot x[m \cdot H + n],$$

where $w[n]$ is the window function, $m$ is the frame index, and $H$ is the frame shift.

In Figure 3, a fast Fourier transform is then applied to each window to convert the time-domain signal into a spectrogram:

$$X[m, k] = FFT\{x[m]\}.$$

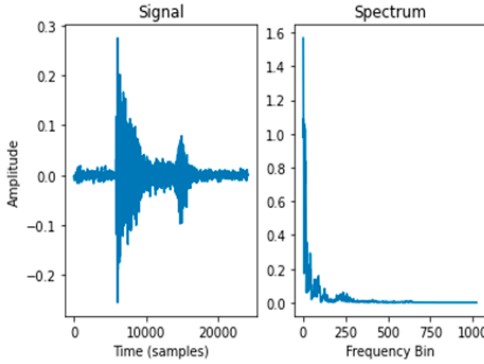

**Figure 3.** Signal and spectrum of audio.

Then in Figure 4, the energy of the Mel filter bank is calculated by multiplying the spectrum with the frequency response of a set of Mel filters:

$$S[m, i] = \sum_{k=0}^{N-1} |X[m, k]|^2 \cdot H[i, k],$$

where $N$ is the FFT points, $H[i, k]$ is the frequency response of the $i$th Mel filter.

In Figure 5, we transform it from log-Mel spectrogram to MFCC by first performing logarithmic transformation to simulate the nonlinear perception of the human ear:

$$M[m, i] = log(S[m, i]).$$

The discrete cosine change for each column gives the MFCC coefficient:

$$C[m, c] = \sum_{i=0}^{M-1} cos\left(\frac{\pi c}{M}\left(i + \frac{1}{2}\right)\right) \cdot M[m, i].$$

where $M$ is the number of Mel filters and $c$ is the index of the MFCC.

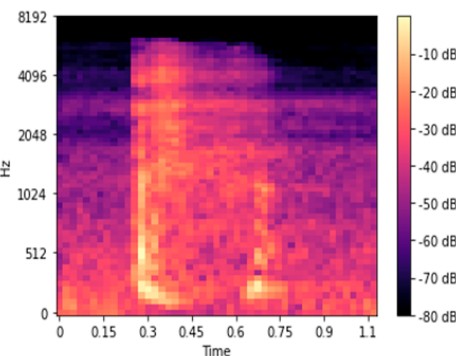

**Figure 4.** Mel-spectrogram of audio.

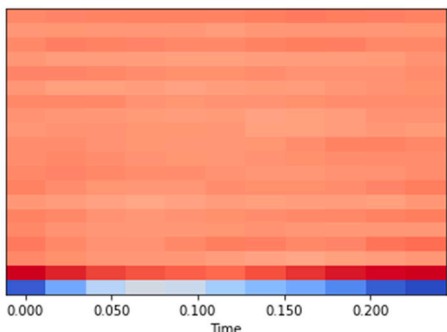

**Figure 5.** MFCC of audio.

The entire process can be represented as a function $D$ operating on the audio waveform $A$ and yielding a matrix of MFCC coefficients:

$$D : A \rightarrow a = \begin{bmatrix} a_{1,1} & \cdots & a_{1,C} \\ \vdots & \ddots & \vdots \\ a_{N,1} & \cdots & a_{N,C} \end{bmatrix}$$

where $N$ denotes the number of frames, and $C$ is the number of MFCC coefficients. Each $a_{i,c}$ represents an MFCC coefficient for the $i$th frame and $c$th coefficient.

After obtaining the labeled audio dataset and converting it into 2D acoustic features, we augment the data to cope with the small amount of data. First of all, according to the basic characteristics of the audio, we scaled its pitch, tone, etc. in equal proportions and carried out noise reduction processing to reduce the noise for emotional judgment. At the same time, in order to expand the data set, we simulate the ambient white noise and add Gaussian noise with different coefficients.

$$Y[k] = IFFT\{X[k] - N[k] - \alpha\}$$

where $Y[k]$ is the denoised signal in the frequency domain, $X[k]$ is the noisy signal in the frequency domain, $N[k]$ is the estimated noise signal in the frequency domain, $\alpha$ controls the amount of noise reduction and *IFFT* denotes the Inverse Fast Fourier Transform.

Data augmentation for a single piece of audio can be expressed as follows:

$$T' = T \times 2^{(n/12)},$$

$$R' = R \times r,$$

where $T'$ denotes the adjusted time axis, and $T$ is the original time axis, and $n$ is the number of semitones. Additionally, $R'$ is the adjusted sampling rate, $R$ is the original sampling rate, and $r$ is the stretching factor, representing the stretching operation.

Gaussian noise addition involves introducing randomly generated values from a Gaussian distribution to an audio signal. The process can be expressed as:

$$X' = X + N$$

where $X'$ and $X$ denotes the adjusted audio signal and the original audio signal, respectively. $N$ is random noises conforming to a Gaussian distribution, which can form with a mean of zero and a standard deviation σ.

### 3.2.2. Structure

In the realm of audio emotion recognition, we use convolutional neural networks (CNNS) which is shown in Figure 6 to capture key features in audio signals. The CNN model includes convolution layer, pooling layer and fully connected layer, in which convolution layer is used for local feature extraction, pooling layer is used for dimensionality reduction, and fully connected layer is used for emotion classification by learning the relationship between features. Activation functions such as the rectified linear unit (ReLU) are employed to introduce nonlinearity, thereby enhancing model expressiveness. In the training process, we use the cross-entropy loss function and adjust the model parameters by optimization algorithms such as gradient descent. To evaluate model performance, we utilize validation sets for model selection while using test sets to quantify model accuracy, recall, and F1 scores. Furthermore, adversarial tests are conducted by introducing audio samples of different emotional expressions to verify the robustness of the model for emotional changes. With this methodology, we aim to build an efficient audio emotion recognition model to provide accurate and reliable emotion classification in practical applications.

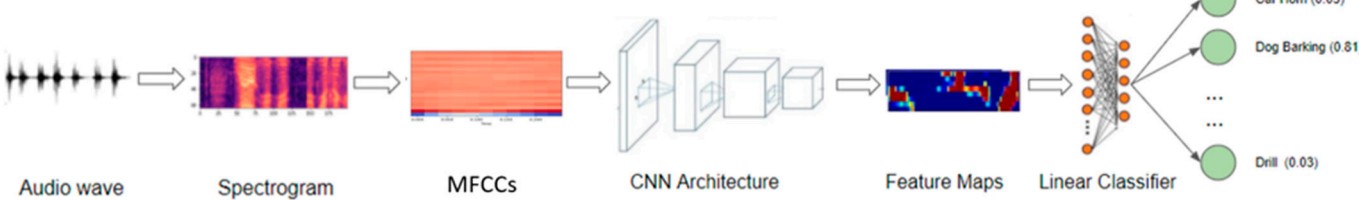

**Figure 6.** Architecture of Convolutional Neural Networks.

As for the convolutional layer, we chose three different sizes which are 3, 4, 5 and separately trained the model. Since we have obtained a matrix of MFCC coefficients with dimensions $N \times C$, we defined a new matrix X with dimensions $(N - K + 1) \times (K \times C)$. Each of X corresponds to a 'window' of $K$ adjacent MFCC coefficients across all coefficients for a particular frame. The matrix is then reshaped into a 3D $X'$ with shape as $(N - K + 1) \times C \times K$, which can be used as input to a CNN. Each slice along the first dimension represents a "frame" or input sample for the CNN, and the second and third dimensions capture the local structure of adjacent MFCC coefficients within each frame.

### *3.3. Graph Convolutional Network*

### 3.3.1. Graph Construction

We construct the corresponding graph $G_i = (V_i, E)$ for a given speech signal $x_i$, where $V$ represents the set of nodes and $E$ represents the set of edges between all nodes. The adjacency matrix $A$ of the graph represents the connection relations and weights between nodes. Our construction strategy is to divide the speech signal into $N$ overlapping short segments (frames), each corresponding to a node in the graph. Each node is concatenated in the temporal order of the audio segments to construct a cyclic graph. The special feature of this construction is that it simplifies the subsequent graph convolution operation, whose adjacency matrix is denoted as:

$$A_{i,j} = \begin{cases} 1, & i = j + 1 \\ 1, & i = 0, j = N \\ 0, & else \end{cases}$$

Each node $v_i$ also carries an associated node feature vector xi, which contains low-level descriptors (LLD) extracted from the corresponding speech fragment. All node eigenvectors form an eigenmatrix $X$.

Temporal Dependency Encoder (TDE):

After the cyclic graph has been constructed, in order to compute the initial weights of the edges between nodes, we introduce a Time Dependent Encoder (TDE) in Figure 7 for calculating the temporal dependencies between the nodes. The process of graph construction for longer audio inevitably leads to a segmentation of a relatively complete piece of information, resulting in uniform edge weights that may not provide a complete representation of the correlation between nodes. In response, TDE employs a Long Short-Term Memory (LSTM) model to discern temporal dependencies among audio segments. The model's storage unit retains information from the previous moment, making it more suitable for processing time-series data. Following temporal segmentation of audio features, TDE categorizes nodes exhibiting stronger correlations into cohesive groups, thereby assigning weighted edges in the cyclic graph based on semantic correlations. This initialization of edge weights strengthens the ability of subsequent GCNs to learn inter-segment correlations. The TDE can be represented as:

$$(E_i, C_i) = LSTM(V_i, E_{i-1}, C_{i-1}),$$

where $E_i$ denotes weight of the $i$th edge, which connecting node $V_i$ and $V_{i-1}$. $C$ denotes the state of LSTM cells, which is recurrently updated.

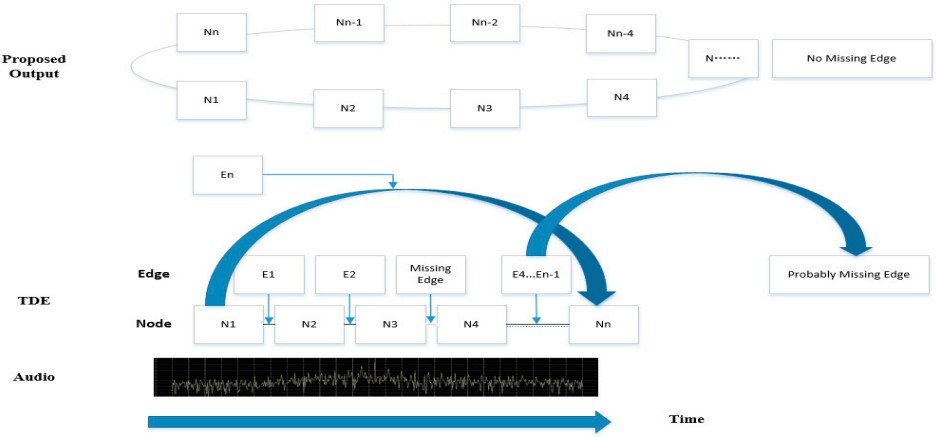

**Figure 7.** Architecture of Temporal Dependency Encoder (TDE).

### 3.3.2. Graph Classification

A graph convolutional network (GCN) is used to learn the relationship between a set of cyclic graphs and their corresponding true labels. The employed GCN consists of two graph convolutional layers and their corresponding pooling layers. A linear classifier is arranged after the last graph convolution layer to predict discrete sentiment labels. Since the node vectors of the initial cyclic graph consist of the acoustic features of the segmented speech segments, and the edge vectors are weighted by the semantic correlations between the speech segments, this allows the GCN to learn the semantic relationships efficiently and quickly between the speech segments in a way that updates the node embeddings and edge embeddings. Since the cyclic graph $G_i$ is constructed from a single audio segment $x_i$, the sentiment classification task is transformed into a graph classification task. We use tiling in order to implement the graph representation and use a linear classifier to predict the sen-

timent labels represented by individual graphs. Similar to the approach used in the CNN branch, the cross-entropy loss function is used to compute the gap between the predicted labels and the true labels to enable updating of the parameters in the GCN. Essentially, the point of the GCN branch is to utilize the temporally relevant speech information contained in the cyclic graph structure to improve the performance of sentiment recognition.

### 3.4. Energy Efficiency

As hardware computing power continues to advance, deep learning methodologies have demonstrated remarkable performance across diverse tasks. Pushing the performance ceiling for each task is an exciting direction, but deep learning is also an increasing resource drain. The EmoNet model we established saves a lot of computing power resources and is more convenient for humanoid emotion recognition.

In order to reduce resource consumption while maintaining performance, we propose an efficient graphic-based hybrid learning model that focuses on speech emotion recognition for humanoid robots. The model amalgamates the advantages of CNNS and graphical data to achieve greater energy efficiency. Through the CNN model, we can capture local features of speech data and enhance the extraction of temporal dynamic information. Compared to traditional LSTM models, our model is more lightweight because it reduces the number of training parameters. Building speech data into graphical data not only strengthens the relationship between speech signals, but also diminishes the number of training parameters, which improves energy efficiency and reduces the energy overhead for training and reasoning. This combined energy efficiency boost makes our hybrid learning model more sustainable and environmentally friendly for speech emotion recognition tasks.

### 3.5. Robot Implementation

We mainly introduced four function modules of the humanoid robot. Once a person speaks something, the robot will collect voice information using the microphone. Meanwhile, the sonar module will be able to locate the distance and direction of the person. By turning its head around, the camera will find the matched person based on former data. After comparing all suited information, the robot will identify the specific human speaker.

## 4. Experiments and Results

In this section, we first describe the details of the datasets and our experimental setup. This is followed by our experimental results, analysis and discussion.

### 4.1. Description of Datasets
IEMOCAP Dataset

The IEMOCAP (Interactive Emotional Dyadic Motion Capture) dataset is a comprehensive collection designed for audio-based emotion recognition research. This dataset features recordings of dyadic interactions, capturing both audio and visual data in various scripted and improvised scenarios. Collected in a naturalistic setting, IEMOCAP offers a rich and diverse set of emotional expressions, including happiness, sadness, anger, and neutrality. One notable aspect of the dataset is its multimodal nature, providing researchers access to audio, video, and text transcriptions for a more holistic understanding of emotional dynamics. Additionally, IEMOCAP includes speaker information, facilitating studies on individual differences in emotional expression.

### 4.2. Details of Implementation
4.2.1. Software

The hybrid networks were constructed in the PyTorch framework and Python-3.9 with Intel Core i7 CPU and NVIDIA GTX 1650Ti GPU. Meanwhile, we used the Nao Robot from Softbank Robotics (Issy-les-Moulineaux, France) for humanoid part.

We chose BCELoss as the loss function, Adam as the optimizer, and set the regularization options and dropout. We set the batch size to 32, the weight decay to $1 \times 10^{-5}$, the

initial learning rate to 0.1, and set the early stop policy, defined patience = 5, and stopped training when the loss on the verification set did not improve in five consecutive epochs. The imbalanced-learn library introduces oversampling and undersampling methods to deal with the unevenness of data sets. For oversampling, RandomOverSampler is used, and the sampling strategy is set as 'minority', so that the number of minority samples can reach the number of majority samples. For undersampling, the RandomUnderSampler is adopted, and the sampling strategy is also set as 'majority' to reduce the number of samples of majority categories.

### 4.2.2. Hardware

As can be shown in Figure 8 is the entity diagram of the Nao robot. We mainly use four modules, namely the microphone, speaker, camera and sonars. The microphone module serves the primary function of capturing human voice data, while the camera and sonars contribute to spatial localization and the identification of specific human speakers for targeted interaction. The speaker diagram is the robot's voice system. After the model docking and information transmission are successful, the speaker will output and interact with human beings.

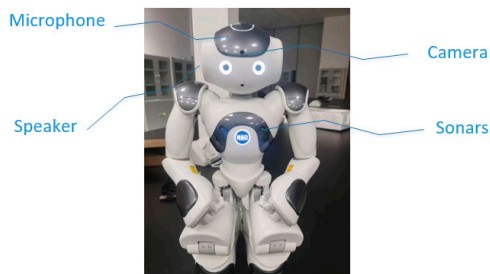

**Figure 8.** Mainly used functional modules on the humanoid robot.

Figure 9 shows the robot operation interface. After the synchronous connection through the ip address, the corresponding functions in the left toolbar and the python script module are used in the interface to carry out human–computer interaction steps.

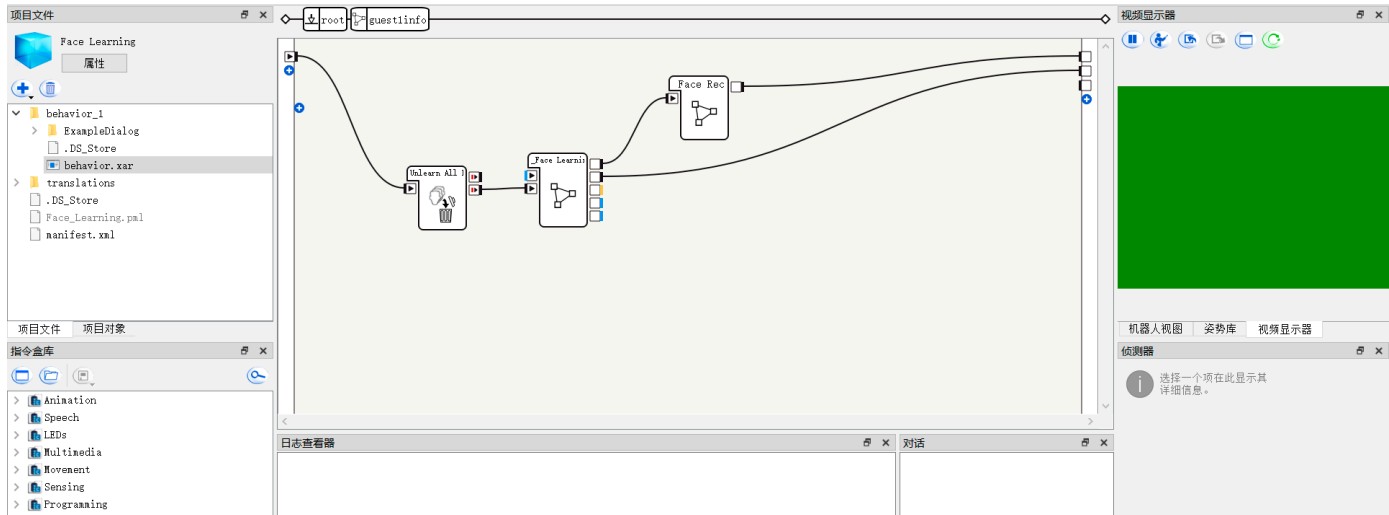

**Figure 9.** Operation Interface of humanoid robot. "项目文件" means project files, "属性" means attributes, "项目对象" means project object, "指令盒库" means instruction box, "日志查看器" means log viewer, "对话" means dialogue, "视频显示器" means video display, "机器人视图"means robot view, "姿势库" means posture library, "侦测器" means detector, "选择一个项在此显示其详细信息" means Select an item to display its details here.

### 4.3. Experiment Results

The architecture of the hybrid model epitomizes an elegant simplicity combined with potent capabilities. A linear layer, activation function, dropout layer, and output layer constitute the backbone of the model. Regularization terms are introduced to enhance its generalization capabilities, a crucial facet for robust performance across diverse datasets. A critical aspect of the model's success lies in its ability to outperform individual CNN and GCN models. The hybrid model orchestrates a harmonious blend, leveraging the unique features extracted by each component. This synergy propels enhanced accuracy, F1 score, and recall, painting a compelling picture of the benefits derived from this fusion strategy.

In the pursuit of broader applicability, regularization mechanisms and early stopping strategies are woven into the model's fabric. Notably, these features contribute to heightened generalization capabilities, allowing the model to discern patterns that transcend the idiosyncrasies of the training data. The strategies of oversampling and undersampling play pivotal roles in addressing class imbalance, a common challenge in real-world datasets. The model, thus fortified, excels in learning from underrepresented classes, resulting in an uplifted overall performance.

### 4.4. Comparison of Different Models

In order to verify the performance improvement and computation resource reduction in the hybrid network over the traditional CNN and GCN, we conducted experiments. The parameters of the EmoNet model and the traditional neural network model were set the same to ensure the constant number of convolutional layers. In tandem, 20 epochs and the early stop function were set to prevent the model from consuming high energy but reducing optimization performance, thus achieving energy efficiency. As shown in Table 1, the data in the table shows the accuracy differences under several models, namely the Hybrid model > CNN model > GCN model under accuracy conditions. Through training, the emotion recognition accuracy of our proposed model on the IEMOCAT dataset is as high as 84%, which is 4.23% higher than the traditional CNN model. Initially, the accuracy of the model is around 81%, which is lower than expectation. Thus, we decided to improve the model in several aspects. Through the adjustment of the learning rate and optimizer like Adam and SGD, we obtained imperceptible changes from 81.6% to 82.1%. Thus, we tended to add the model complexity by increasing the number of hidden layers or nodes and tried a deeper network structure. Upon these experiments, our model achieved an accuracy of about 83% so we decided to combine them together. At last, we increased the training time and used larger batch sizes for conformation, which led to an accuracy of 84.197% of the Hybrid model which is better than before. At the same time, we carried out repeated experiments many times to ensure the stability of the model. When the traditional CNN model ran 20 epochs each time, the average training times of the Hybrid model proposed by us was 13 times, which reduced the computing power requirement. Regarding the GCN model, while it may have exhibited certain limitations in its graph structure compared to previous methodologies utilized with the IEMOCAP dataset, the achieved accuracy remained commendable, and the accuracy rate reached about 64% on the basis of cycle and line connection methods. The graph structure network we built had some shortcomings, but the result was still acceptable.

**Table 1.** Accuracy and evaluation of different models.

| Audio Emotion Recognition | Accuracy(%) | F1 Score | Recall | Precision |
|:---:|:---:|:---:|:---:|:---:|
| CNN | 79.97 | 0.51 | 0.54 | 0.49 |
| GCN | 58.32 | / | / | / |
| Hybrid | 84.20 | 0.84 | 0.83 | 0.85 |

Compared with the three kinds of neural networks, the accuracy of GCN is not excellent. We speculate that GCN has a remarkable effect on the image construction ability

of pixel nodes, but its ability to capture audio data waveform is not outstanding. In the same case, the CNN model has a strong ability to extract audio spectra and MFCC features, making the gradually improved CNN model better than the GCN model in accuracy. On this basis, the Hybrid model takes advantage of the advantages of the CNN model, and in the convolutional layer, extracts the abstract feature vectors of the two models and merges them accordingly. By combining the capture of audio details by CNN and the graph structure process of GCN, the noise and other special values in the sound characteristics are screened and de-cluttered, which improves the accuracy of the model.

By comparing the training loss and test loss of the model in Figure 10, we know that the CNN model is in a rapid decline stage and continues to decline with the progress of training, because every training session will reduce the existing loss to optimize the accuracy of the model, while the verification loss is consistent with it at first, and then maintains a relatively moderate percentage. This is because the model has no predictive ability at the initial stage of training, but after the training iteration, the predictive effect is gradually significant and stable. Conversely, the Hybrid model capitalizes on pre-trained feature vectors, resulting in superior training and prediction outcomes from the onset. In the new model, the loss value generated is low and stable. The following two diagrams in Figure 11 show the improvement of the model accuracy with the epoch. You can see that both are fluctuating and rising while the Hybrid model tends to flatten out and produces a higher final result.

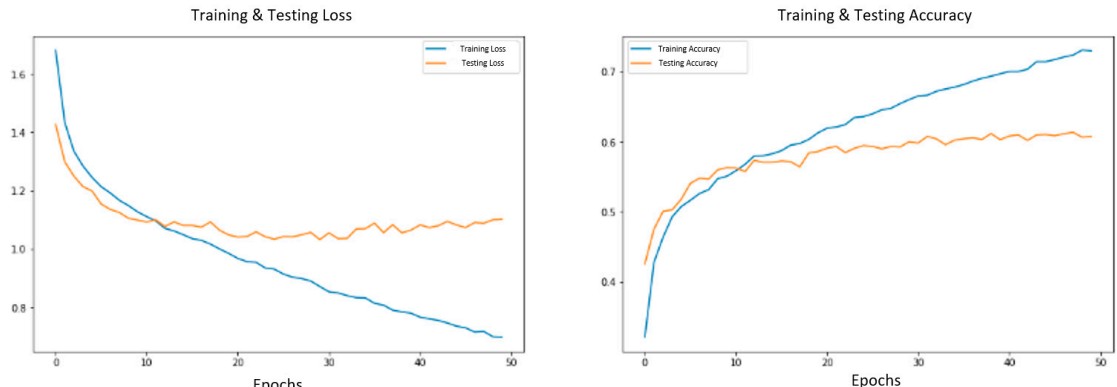

**Figure 10.** Loss output of CNN and Hybrid models.

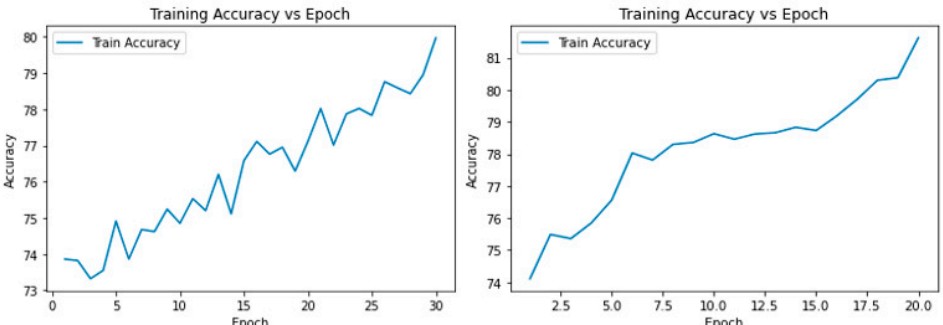

**Figure 11.** Accuracy curve with different epochs.

### 4.5. Performance of Hybrid EmoNet Model

In the evaluation of our model's performance, the ROC curve in Figure 12 exhibits a notable discriminative capacity, particularly evident at a threshold of 0.1, where a commendable true positive rate of approximately 0.7 to 0.8 is achieved alongside a relatively low false positive rate. As the threshold increases, the true positive rate tends towards 1, indicative of heightened model conservatism and potentially enhanced precision at the expense of recall. The overall smoothness and trajectory towards the top-left corner of

the ROC curve imply a robust trade-off between sensitivity and specificity across various thresholds. Furthermore, the quantitative assessment through the AUC-ROC score aligns with these observations, substantiating the model's strong overall performance.

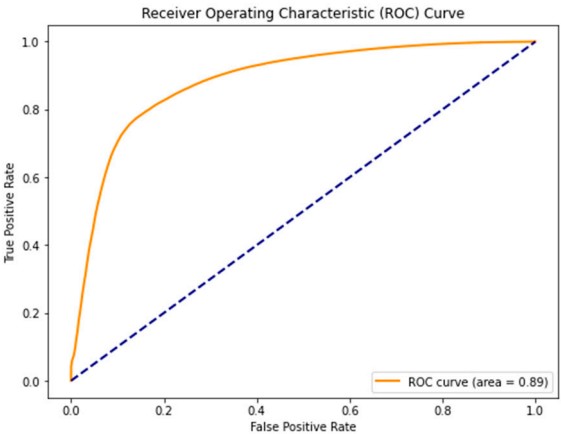

**Figure 12.** ROC curve of Hybrid model.

### 4.6. Implementation on Humanoid Robot

As for Figure 13, it shows the realization of streaming media transmission between computer and robot. In this study, we adopted a distributed programming strategy to achieve remote access to Naoqi SDK functions by running a Python 3.9-based client locally and connecting to a robot running a Python 2.7 environment. This client sends a request to the robot environment to obtain the robot's perceptual data, such as images or sensor readings. By taking the form of a data stream, we effectively transmit the information observed in the robot environment back to the machine. This information transfer involves communication across Python versions, ensuring compatibility and effectiveness in different environments. After the data arrived at the machine, we extracted and saved the audio data, showing a successful example of robot data acquisition and local processing in a heterogeneous Python environment.

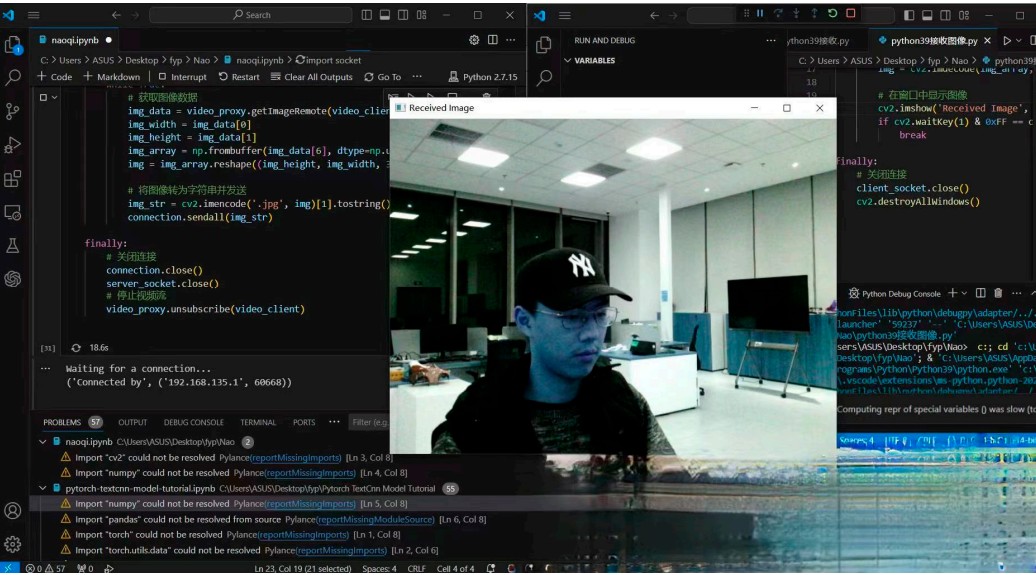

**Figure 13.** Realization of communication between computer and humanoid robot. "获取图像数据" means get image data, ""将图像转为字符串并发送" means convert the image to a string and send it, "关闭连接" means closing connection, "停止视频流" means stop the video stream, "在窗口中显示图像" means display the image in the window, "关闭连接" means close connection.

As can be shown in Figure 14, it communicates with the interactive interface for emotion recognition-connecting robot using Choregraphe (version 2.8.6.23). In the core stage of this research, we focus on achieving accurate recognition of robot emotions. By using deep learning techniques, we trained an emotion recognition model in a Python 3.9 environment and successfully compressed it for deployment in a Python 2.7 environment through efficient transformation methods. The emotion recognition model takes tone as the main feature, and can accurately judge the emotional states recognized by the robot in its interaction with humans.

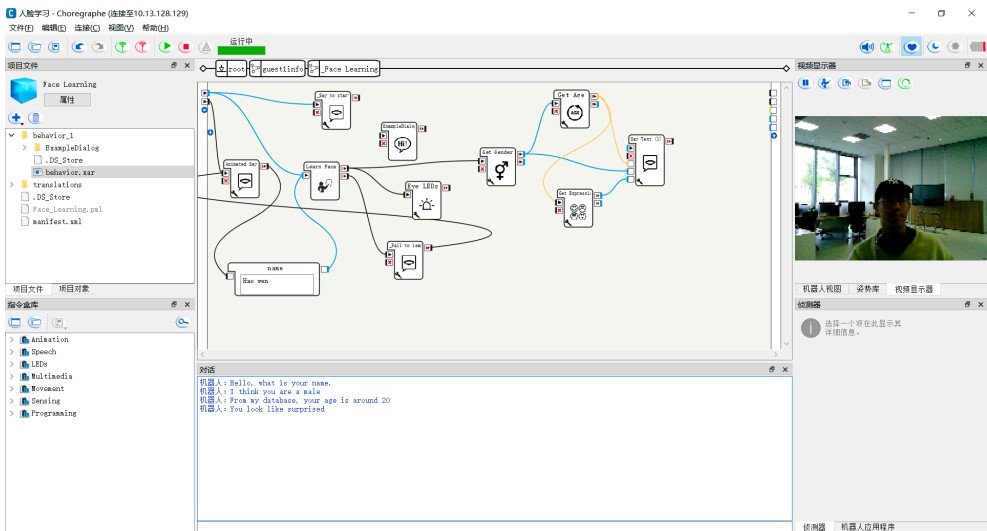

**Figure 14.** Realization of humanoid robot emotion recognition using Choregraphe. "项目文件" means project files, "属性" means attributes, "项目对象" means project object, "指令盒库" means instruction box, "日志查看器" means log viewer, "对话" means dialogue, "视频显示器" means video display, "机器人视图" means robot view, "姿势库" means posture library, "侦测器" means detector, "选择一个项在此显示其详细信息" means Select an item to display its details here. "人脸学习" is the file that I name, which means learning face, the "文件,编辑,连接,视图,帮助" respectively means file, compile, connect, version, help", "运行中" means running.

As shown in Figure 15, after going through the trained model, through the pre-designed reaction steps, the robot will react differently to the judgment of different emotions; for example, a different posture, covering the eyes, spreading the hands, etc. In addition, they respond by saying different words in preset terms.

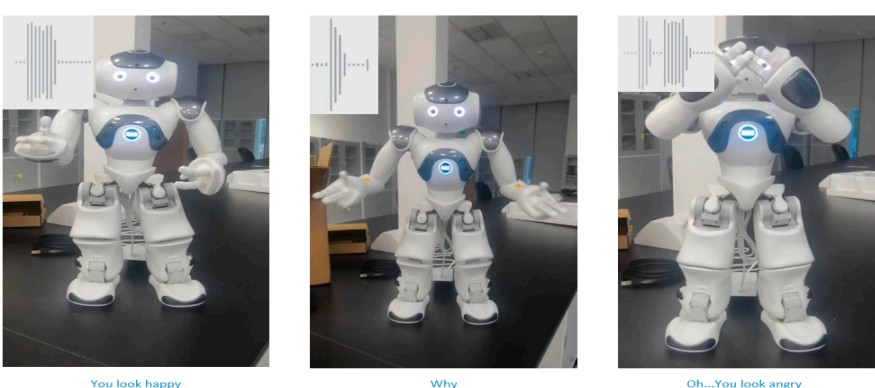

**Figure 15.** Reaction of humanoid robot on emotion recognition.

### 4.7. Theoretical Applications in Other Modalities

Based on the above experimental evaluation results on the audio data set, we propose the performance and potential of this hybrid model for emotion recognition in other

modes. In the field of emotion recognition, audio, image and text are the three fundamental analysis modes, which are the basic elements that can best judge each other's emotions in human daily life. The performance of the mixed model in audio emotion recognition indirectly explains its feasibility in the other two modes mentioned, especially in the visual aspect. Based on the strong GCN model for image data analysis, the hybrid model combined with the two is likely to be significantly improved when the accuracy is further improved. In addition, the video streaming data observed by the robot is still 2D images, but in real life it is mostly 3D image data. Using the CNN model to process text and audio information and GCN model to process 3D image information, the possibility of constructing a multi-angle and multi-modal mixed model for emotion recognition has also been significantly improved.

*4.8. Limitation of Experiment*

Since the model is only a single mode model, the experiment has great limitations. First of all, for human beings, the way to express emotions is not limited to voice. Actions and texts are also means to detect emotions. People can say surprising things in extremely calm voices, which causes the model to fail in identifying emotions. Secondly, in a noisy environment, it is difficult for the robot to react quickly according to the audio source, so that the audio information collected is biased, or the audio information of too many people is collected at one time. For such audio data, it is difficult to separate the tracks, which causes the model to not work properly.

The presence of noise significantly affects the accuracy of Speech Emotion Recognition (SER) systems. Noise introduces distortions into the speech signal, obscuring important emotional cues and complicating the recognition process. A high-intensity noise can mask subtle variations in speech patterns associated with different emotions, making it challenging for models to accurately identify emotional states. Additionally, noise-induced signal degradation may lead to information loss, particularly in the high-frequency range where crucial emotional cues are often encoded. Furthermore, noise may introduce misleading features that models might erroneously interpret as emotional signals, leading to incorrect classification outcomes. Moreover, if SER models are trained on noisy data, they may become overly sensitive to noise patterns, resulting in decreased performance in real-world scenarios where noise levels vary. Overall, noise impacts accuracy in SER by degrading signal quality, masking emotional cues, inducing information loss, causing misclassification, and potentially leading to overfitting and decreased robustness. Effective noise mitigation strategies and robust model architectures are essential for improving accuracy in noise-affected SER applications.

## 5. Discussion

In the field of emotion recognition, there have been many models to make predictions. From the earliest basic neural network model, such as the Random Forest model, Gharsalli [34] obtained 76.7% accuracy when the decision tree was set to 400 after several experiments. Although it has reached a rather high level, compared with our model, there are still shortcomings. Similarly, in 2020, Yu [35] used the Attention-LSTM-Attention model to predict emotion, and the highest accuracy rate obtained was 67.66% (up or down 3.4%), which could not surpass the performance of our model even at the highest value. With the further improvement, the CNN model also gradually emerged. After a series of optimizations, Meyer's model obtained a very satisfactory result, with an average prediction accuracy of 76.9% and a fluctuation of 7.9%. Theoretically speaking, the highest value can exceed most models. As a result, the drawbacks of the model are obvious.

However, despite advances in models such as Random Forest, Attention-LSTM-Attention, and CNN, they still face certain limitations when it comes to emotion recognition. Because Random Forest models rely on decision trees, it may be difficult to capture nuances in emotional data. Similarly, while the attention-LSTM-attention model combines attentional mechanisms to focus on relevant parts of the input sequence, it may not be effective in deal-

ing with long-term dependencies and complex temporal dynamics in emotional signals. On the other hand, CNN models may lack the ability to capture sequential dependencies and may require a lot of data preprocessing to effectively extract relevant features.

In contrast, our fusion model leverages the strengths of various architectures and technologies, such as convolutional layers, pooling mechanisms, and fully connected layers, to comprehensively analyze the graph structure of speech signals and extract meaningful emotional features. By integrating these components, our models achieve superior performance in emotion recognition tasks, going beyond the limitations of a single model and providing a more robust and accurate solution for capturing and interpreting emotional content in speech data.

In addition, from the perspective of various emotion analyses of human–computer interaction, as emotion recognition research continues to evolve, our fusion model demonstrates the potential of integrating multiple approaches to achieve more comprehensive and effective solutions in this field.

Our findings have far-reaching implications for various fields, especially in the areas of human–computer interaction (HRI) and social intelligence. By improving the ability of emotion recognition models, our work lays the foundation for more intuitive interactions between humans and robots. In the context of HRI, the ability of robots to understand and respond to human emotions is critical, and our model builds a bridge for this communication, further improving the possibilities.

In addition, when it comes to social intelligence and emotional computing, interpreting human emotions with greater accuracy and sensitivity recognition, our model opens the way for innovative solutions in areas such as healthcare, education, and customer service. For example, SoftBank has used Nao robots to teach young children. After identifying the emotions of each child in a multi-crowd state, the robot can develop personalized responses for different people. In terms of customer service, after listening to customer calls, AI customer service can obtain customer demands through emotion analysis and judge their satisfaction. Together, by harnessing the power of advanced computational models, we are paving the way to integrate technology into our daily lives, making it more emotionally intelligent and empathetic, ultimately enriching the human experience and creating deeper connections between humans and machines.

## 6. Conclusions and Future Work

In conclusion, our study introduces a novel HybridEmoNet model, seamlessly integrating Convolutional Neural Networks (CNNs) and Graph Convolutional Networks (GCNs) for audio emotion recognition within humanoid robots. The proposed model exhibits superior performance compared to individual CNN and GCN models, achieving an accuracy of 81.61% on the IEMOCAP dataset. The fusion of CNN's spatial cues and GCN's structural insights results in a harmonious blend, effectively capturing nuanced emotional patterns. Additionally, the model demonstrates energy efficiency, reducing computational resource requirements while delivering competitive accuracy. The experiments, conducted on the IEMOCAP dataset, highlight the model's robustness, adaptability, and potential applicability across various modalities beyond audio, laying the groundwork for advanced emotion recognition systems in humanoid robots. This research contributes to the evolving landscape of emotion-aware artificial intelligence, with implications for human–robot interaction, social intelligence, and beyond.

While GCN-based hybrid models have shown significant improvements in emotion recognition compared to traditional methods, there still exists some limitations. First of all, our feature fusion method is relatively simple. The feature vectors of the two models are tiled into one dimension for concatenation, and then the fully connected layer and softmax are used for further fusion. Therefore, more general fusion methods will be explored in the future. Secondly, we have optimized part of the GCN model, but we have not adopted the optimization measures in feature extraction for the CNN model. After understanding the attention mechanism and adaptive connection, we may draw up a new layer to intro-

duce these two points and enhance the model's discrimination for different categories. In addition, only one mode of audio is used for emotion recognition in this paper, and in the current environment, multi-modal model has become the norm. We will also expand this model to other modes such as image and txt for fusion analysis.

**Author Contributions:** Conceptualization, H.W., K.P.S. and H.X.; methodology, H.W., H.X. and K.P.S.; resources, K.P.S.; data curation, H.W., H.X., K.P.S. and J.C.; writing—original draft preparation, H.W., H.X. and K.P.S.; writing—review and editing, K.P.S., J.C. and L.M.A. All authors have read and agreed to the published version of the manuscript.

**Funding:** This research received no external funding.

**Data Availability Statement:** The research dataset is IEMOCAP. Link: https://paperswithcode.com/dataset/iemocap (accessed on 3 July 2023).

**Acknowledgments:** The first author Haowen Wu and corresponding author Jasmine Seng Kah Phooi would like to express our sincere gratitude to Austin Fu from Softbank Robotics China who has provided the technical support and training for the Nao platform to be used in this research project. His dedication, expertise, and commitment were instrumental in the realization of our research objectives. We are thankful for him valuable insights, collaborative spirit, and unwavering support throughout the research project.

**Conflicts of Interest:** The authors declare no conflicts of interest.

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
