# Peer review of "Energy Efficient Graph-Based Hybrid Learning for Speech Emotion Recognition on Humanoid Robot"

_electronics, doi:10.3390/electronics13061151_

Round 1

Reviewer 1 Report

Comments and Suggestions for Authors

This paper presents a novel deep graph-based learning technique for speech emotion recognition which has been specifically tailored for energy efficient deployment within humanoid robots.

The paper overall is good

I have two comments

1- A section for discussion after results would be better and try to compare more with other works

2- an accuracy  of 81% for Hybrid is that much and I believe you can do more here in the data or model to have better percentage

Regards

Comments on the Quality of English Language

please improve the level of english in this paper

Author Response

Please refer to the response statement for Reviewer 1. Thanks

Reviewer 2 Report

Comments and Suggestions for Authors

The authors present a deep graph-based learning technique for speech emotion recognition tailored for humanoid robots, focusing on energy efficiency and reduced computational complexity. The chosen title aptly reflects the paper's content, and the abstract is clearly articulated.

In the introduction, the basis of the research is established through a discussion of existing literature on speech emotion recognition using deep learning techniques. While a research gap is identified, the manuscript lacks a dedicated literature review section. It is recommended that the authors include a specific section titled "Literature Review" or "Background." This section should elaborate on the research mentioned in the introduction, further dividing the discussion into areas of sound emotion recognition and deep learning techniques. This addition would enhance the manuscript's structure by providing a comprehensive overview of the field and strengthen the manuscript's context and contribution.

The methodology, experiment and result section are explained well (with some exception as mentioned below).

Line 222-224: "basic characteristics of the audio, we scaled its pitch, tone, etc in reducing noise equal proportions and carried out noise reduction processing to reduce the noise for emotional judgment" - These sentences needs further explanation as to what "etc in reducing noise equal proportions" could mean. Another problem with these sentence is that noise reduction is just explained here and it does not incorporate the necessary steps(/techniques) required to perform noise reduction for the speech data ( a simple equation or sentence would suffice).

The approach of using a hybrid CNN and Graph based model is interesting. In the Table 1. Accuracy and evalutaion [minor typo Line 400] of different models, the data shows the hybrid model has an accuracy of 81.61% as compared to CNN and GCN but no where the authors have mention where the model was unable to perform or under what circumstance the model was unable to detect emotion. The discussion of how noise was impacting the accuracy, or different factors that could impact the accuracy needs to be discussed further.

Figures:

Figure 8. Mainly used functional modules on the humanoid robot mentions microphone, speaker, camera, sonars. The authors have explain that camera is used for language positioning and identifying the specific human speaker for special capture. But no where in the methodology this has been explained in detailed.  

Figure 9. Operation Interface of humanoid robot - too small

Figures 13 - 15.

Fig 13. Realization of communication between computer and humanoid robot

Fig 14. Realization of humanoid robot emotion recognition using Choregraphe

Fig 15. Reaction of humanoid robot on emotion recognition

These images have not been referenced properly. A suggestion would be showing at which stage these images are related to. 

Typos: 

Line 402 - DCN (not sure if this should be GCN)

Line 408 -  ">CNN" model ">GCN" (not sure if these symbols are purposely used ">" , if there is a reason, suggestion would be provide reasoning)

Author Response

Please refer to the response statement for Reviewer 2. Thanks

Reviewer 3 Report

Comments and Suggestions for Authors

The research contributes significantly to the evolving landscape of emotion-aware artificial intelligence, particularly in the context of humanoid robots. Emphasizing the potential implications of findings for human-robot interaction, social intelligence, and other relevant domains would further underscore the significance of work. 

The conclusions show the contributions and implications of the research while providing clear directions for future investigation. Addressing the aforementioned points would enhance the clarity, rigor, and impact.

Author Response

Please refer to the response statement for Reviewer 3. Thanks

Round 2

Reviewer 1 Report

Comments and Suggestions for Authors

Great work

Comments on the Quality of English Language

Great work

Reviewer 2 Report

Comments and Suggestions for Authors

Thank you for your revised manuscript. The authors have addressed all the concerns. This version is ready to be accepted.